# Leptomeningeal Carcinomatosis: A Clinical Dilemma in Neuroendocrine Neoplasms

**DOI:** 10.3390/biology10040277

**Published:** 2021-03-28

**Authors:** Leonidas Apostolidis, Jörg Schrader, Henning Jann, Anja Rinke, Sebastian Krug

**Affiliations:** 1Department of Medical Oncology, National Center for Tumor Diseases, University Hospital Heidelberg, 69120 Heidelberg, Germany; Leonidas.Apostolidis@med.uni-heidelberg.de; 2I. Medical Department—Gastroenterology and Hepatology, University Medical Center Hamburg-Eppendorf, 20246 Hamburg, Germany; jschrader@uke.de; 3Department of Gastroenterology and Hepatology, Charité—University Medical Center Berlin, Campus Virchow Klinikum and Charité Mitte, Augustenburger Platz 1, 13353 Berlin, Germany; henning.jann@charite.de; 4Department of Gastroenterology and Endocrinology, University Hospital Marburg, Baldinger Strasse, 35043 Marburg, Germany; 5Clinic for Internal Medicine I, Martin-Luther University Halle/Wittenberg, Ernst-Grube-Straße 40, 06120 Halle, Germany

**Keywords:** leptomeningeal carcinomatosis, brain metastasis, neuroendocrine tumor, neuroendocrine carcinoma, prognosis

## Abstract

**Simple Summary:**

In four neuroendocrine tumor (NET) centers, we structurally assessed the involvement of the central nervous system with a focus on leptomeningeal carcinomatosis (LC) in patients with neuroendocrine neoplasms. We precisely evaluated the clinical pathological criteria, symptoms, therapy, and outcomes. LC is associated with a poor prognosis but not attributed to special patient characteristics. There is currently an unmet medical need for an optimal treatment strategy for neuroendocrine neoplasm (NEN) patients with LC. In some cases, MRI of the brain and spine might be implemented in the diagnostic workup.

**Abstract:**

Central nervous system (CNS) involvement by paraneoplastic syndromes, brain metastases, or leptomeningeal carcinomatosis (LC) in patients with neuroendocrine neoplasms (NEN) has only been described in individual case reports. We evaluated patients with LC in four neuroendocrine tumor (NET) centers (Halle/Saale, Hamburg, Heidelberg, and Marburg) and characterized them clinically. In the study, 17 patients with a LC were defined with respect to diagnosis, clinic, and therapy. The prognosis of a LC is very poor, with 10 months in median overall survival (mOS). This is reflected by an even worse course in neuroendocrine carcinoma (NEC) G3 Ki-67 >55%, with a mOS of 2 months. Motor and sensory deficits together with vigilance abnormalities were common symptoms. In most cases, targeted radiation or temozolomide therapy was used against the LC. LC appears to be similarly devastating to brain metastases in NEN patients. Therefore, the indication for CNS imaging should be discussed in certain cases.

## 1. Introduction

Leptomeningeal carcinomatosis (LC), also known as carcinomatous meningitis or leptomeningeal metastasis, is defined as spread of the disease to the meninges surrounding the brain and the spinal cord and a rare but frequently devastating complication of advanced cancer. LC is diagnosed in approximately 5% of patients with advanced cancer [1,2]. However, in autopsy studies, it can be detected in up to 20% of cases [3]. LC is most frequently associated with breast cancer, lung cancer, melanoma, gastrointestinal cancer, and cancer of unknown primary [1,2,4]. On the other hand, meningeal involvement can also be detected in primary brain tumors and hematologic malignancies. Prognosis of LC is poor, with median survival ranging from 4–6 weeks when untreated to 2–3 months when treated [5,6]. Current treatment strategies include radiotherapy, systemic antineoplastic agents, and intrathecal chemotherapy.

Neuroendocrine neoplasms (NEN) are a heterogeneous group of malignancies with features of neuroendocrine differentiation which can arise in almost any organ system. Most commonly, the primary is located in the lung or the gastrointestinal tract [7]. NEN are subdivided into well-differentiated low- to intermediate-grade neuroendocrine tumors (NET), also called carcinoids in specific organs such as the lung, and poorly differentiated high-grade neuroendocrine carcinomas (NEC) of small-cell or large-cell type. 

Central nervous system involvement is quite commonly diagnosed (prevalence 2–10%) in small-cell lung cancer (SCLC) [8,9], and only rarely in other NEN [10]. Most published works have focused on parenchymatous brain metastases, and LC in NEN is only very rarely reported. Case series of small cell lung cancer show a LC prevalence of 2% [9]. For other NEN, LC has been reported on a single-case basis of poorly differentiated NEC [11,12,13,14,15,16,17,18,19,20,21], most commonly in large-cell NEC of the lung, Merkel cell carcinoma, and primaries in the head and neck region. For well-differentiated NET, only two cases have been reported so far, both for pancreatic NET [22,23]. Furthermore, several relatively indolent cases of leptomeningeal spread in paraganglioma and pheochromocytoma have been described, mainly in intraspinal primary tumors [24,25,26]. The aim of this study is to analyze the frequency of LC in NEN, as well as clinical characteristics, diagnostic steps, treatment strategies, and outcomes.

## 2. Materials and Methods

All patients with neuroendocrine neoplasm and suspected or proven LC were included in this evaluation. A confirmed LC was present if a liquor puncture had diagnosed malignant cells corresponding to the underlying disease. The suspicion of LC was either based on radiological findings and the presence of clinical symptoms or clinical symptoms alone, excluding other differential diagnoses such as brain metastases. Radiological signs were defined as multiple masses within the subarachnoid space or diffuse leptomeningeal enhancement. The inclusion was independent of the primary tumor. However, patients with small-cell or large-cell lung cancer were excluded from our study. The patients were identified via center-based databases, and the available essential information was extracted and evaluated across centers. The following German centers participated: Marburg, Heidelberg, Hamburg, and Halle (Saale). As comparative cohorts, patients with neuroendocrine neoplasms plus brain metastases (already published [10]) and a collective of metastatic neuroendocrine carcinomas without brain metastases were confronted. Our hypothesis was that patients with neuroendocrine neoplasms and LC have a similarly poor prognosis as patients with metastatic poorly differentiated neuroendocrine carcinomas or patients with neuroendocrine neoplasms and brain metastases. This study was conducted in accordance with the Declaration of Helsinki. 

Statistical analysis was performed using IBM SPSS Statistics. Kaplan-Meier analyses of overall survival and survival since diagnosis of leptomeningeal carcinomatosis were investigated. We used the log-rank test to detect statistically significant differences between groups. Significance was defined as *p* < 0.05. 

## 3. Results

### 3.1. Patient and Tumor Characteristics

In total, we identified 17 patients with neuroendocrine neoplasms and leptomeningeal carcinomatosis (LC) (Table 1). The mean age at the time of diagnosis was 55 years (range 27–73). The group comprises nine male (52.9%) and eight female (47.1%) patients. The majority of patients (n = 14, 82.4%) had poorly differentiated neuroendocrine carcinomas. The distribution according to Ki-67 in the G3 group yielded the following results: five patients (29.4%) with Ki-67 ≤55% and eight patients (47.1%) with Ki-67 >55%. Primary sites were pancreatic (n = 3, 17.6%), lung (n = 3, 17.6%), gastrointestinal (n = 2, 11.8%), cervix and prostate (one patient for each localization), and seven patients with unknown localization (41.2%).

### 3.2. Latency First Diagnosis to Leptomeningeal Carcinomatosis

Median time from initial diagnosis of neuroendocrine neoplasm until diagnosis of leptomeningeal carcinomatosis was 7 months (95% CI 2.9–11.0 months, Figure 1A). The mean age at first diagnosis was 55 years and ascended to 57 years when leptomeningeal carcinomatosis was diagnosed. In only two patients, leptomeningeal carcinomatosis was detected at the initial diagnosis (11.8%). In four patients, the latency period before leptomeningeal carcinomatosis occurred was longer than 24 months. Among them, there were two patients with a G2 (pancreas and atypical carcinoid of the lung) and two patients with a G3 (prostate and cervix) neoplasm. Furthermore, in five patients, leptomeningeal carcinomatosis was confirmed by liquor puncture (29.4%). In the other patients (n = 12, 70.6%), radiological signs of leptomeningeal carcinomatosis and typical neurological symptoms without any signs of brain metastases led to the tentative diagnosis. 

### 3.3. Specific Symptoms in Patients with Leptomeningeal Carcinomatosis

In patients in whom leptomeningeal carcinomatosis was diagnostically confirmed or suspected, a wide spectrum of symptoms was observed (see Table 2). Most complaints were motor or sensory deficits, including paraparesis and paresthesia (each n = 3, 17.6%). Nausea/vomiting and headache as a nonspecific symptom occurred infrequently (n = 1, 5.9%). Impaired vision or loss of visual acuity was apparent in three patients (17.6%). Changes in vigilance, cranial nerve impairment, and nystagmus were summarized under the item ‘Others’ (n = 5, 29.4%). No specific symptoms occurred in two patients (11.8%).

### 3.4. Tumor Stage at Diagnosis and Localization of Distant Metastases

In 15 of the 17 patients (88.2%), distant metastases beyond leptomeningeal carcinomatosis were present at diagnosis. In addition, one patient was diagnosed with leptomeningeal carcinomatosis within 4 weeks after initial diagnosis. Moreover, one patient showed no further tumor manifestation besides the leptomeningeal carcinomatosis. Twelve patients (70.6%) developed multiple localizations (>2) of distant metastases. The most frequent site was lymph nodes (10/17; 58.8%), followed by liver (9/17; 53.9%) and bone metastases (9/17; 53.9%). In nine patients, solid and definable brain metastases (10/17; 58.8%) were found parallel to the leptomeningeal carcinomatosis. Further localizations included lung/pleural cavity (4/17; 23.6%) and others (4/17, 23.6%) such as adrenal, subcutaneous, and spleen metastases.

### 3.5. Treatment and Outcome Data

Local radiotherapy of the affected meninges and, in some cases, whole brain radiation was performed in 11 patients (64.7%, Table 1). In two cases, an intrathecal application of methotrexate was applied (11.8%). Five patients (29.4%) received a temozolomide-based chemotherapy. All other patients received symptomatic treatment and other additional systemic chemotherapy protocols for their underlying disease.

The cumulative median overall survival of the entire cohort was 16 months (95% CI 12.3–19.7). After leptomeningeal carcinomatosis was diagnosed, median overall survival (mOS) was 10 months (95% CI 0–24.0). The 2-year survival rate was calculated with 11.8%. One patient with atypical carcinoid of the lung presented a long-term survival of 42 months. Patients with NEC G3 Ki-67 >55% showed very prompt decease within 2 months after diagnosis of LC in comparison to patients with Ki-67 <55% (Figure 2). Radiological features and treatment of a case with solitary meningeal manifestation is summarized in Figure 3.

## 4. Discussion

Whereas in some malignancies, leptomeningeal carcinomatosis (LC) is not an uncommon tumor manifestation, in neuroendocrine neoplasms (NEN) there are only few individual case reports. As summarized in Table 3, most reported cases are poorly differentiated NEC, with primaries in the lung, uterine cervix, and skin (Merkel cell carcinoma), as well as primary with locoregional meningeal contact such as the sinonasal region, pituitary carcinoma, and intraspinal paraganglioma. In those reports, overall survival after LC diagnosis varied greatly, ranging from several weeks up to more than 12 years, with a median of 4.9 months. However, survival and follow-up data were missing in most cases. 

To date, our case series of 17 patients therefore represents the most structured presentation and largest evaluation of affected patients. Our data show that LC can occur in NEN of different primaries. Patients with GEP-NEN and unknown primary, as well as urogenital and atypical bronchial carcinoids, were involved. Usually, the incidence differs among patients with breast or lung cancer and melanomas between 3–30% [27,28,29]. Clinical symptoms are often unspecific and comprise a wide clinical spectrum. Certainly, the most important differential diagnoses are brain metastases (BM) or paraneoplastic syndromes (PNS) in patients with malignant diseases. In NEN, the risk for BM is low, and the occurrence of PNS is extremely rare [10,30]. Moreover, the radiological diagnosis is often challenging, and cerebral imaging is not routinely included in the work-up of NET or neuroendocrine carcinomas (NEC) [31]. If LC is suspected, the combination of MRI with gadolinium for brain and spine and assessment of the cerebrospinal fluid (CSF) via lumbar puncture (LP) is recommended [32]. When LP is negative but a high clinical suspicion is present, a second collection can increase sensitivity of the cytology. Interestingly, new approaches have been introduced recently to improve the diagnostic accuracy of CSF. Circulating tumor DNA (ctDNA) might be promising but is not ubiquitously accessible [33,34]. Our patient collective also reflects the difficulty of accurately diagnosing LC. 

Additionally, the prognosis remains sobering when LC is diagnosed. The overall survival of our cohort from LC diagnosis was 10 months. LC in NEC with Ki-67 higher than 55% was devastating. Despite treatment approaches in this group, the median survival was 8 weeks and thus comparable to patients without LC-directed treatment [35,36]. In principle, it is assumed that the blood-CSF barrier is disrupted in patients with LC and that systemic therapy should be sufficient as in other metastatic localizations [36]. The therapeutic procedure is mostly guided by retrospective series and expert opinions because only few randomized and prospective trials exist. Most patients in our series received local radiotherapy or whole brain radiation, which was, in some cases, caused by synchronous brain metastases. Methotrexate (MTX) was administered intrathecally only in two patients, as it is the most commonly applied regimen for LC in solid tumors [37,38]. Other commonly applied schedules of intra-CSF therapy include cytarabine, liposomal cytarabine, or thioTEPA (thiotriethylenephosphoramide), however, with only limited experience in solid tumors [39,40,41]. For metastatic NEC temozolomide in combination with capecitabine has been classified as an effective chemotherapy [42]. In LC, temozolomide has been evaluated given its known impact in glioblastomas and CNS penetration. However, beyond some case reports, only one study tested temozolomide in patients with solid tumors and LC [43]. Only the minority of patients benefited from this approach. This might be related to the fact that most of them had lung cancer or melanomas, which are only marginally sensitive to temozolomide. Recently, an open-label phase 2 trial investigated the intravenous application of the immune checkpoint inhibitor pembrolizumab in patients with LC [44]. In this analysis, almost all patients had breast cancer. Overall, pembrolizumab displayed activity as monotherapy with the study meeting its primary endpoint, with a survival rate of 60% at 3 months. In our series, no patient received an immune checkpoint inhibitor, given the fact that this treatment approach is not approved for NEN and patients with active central nervous system metastases are a common exclusion criterion for clinical trials. 

## 5. Conclusions

Although leptomeningeal carcinomatosis (LC) is an infrequent event in patients with neuroendocrine neoplasms (NEN), there is a current unmet medical need for the optimal treatment strategy of LC in those patients. Given the poor survival after diagnosing LC, in some cases, MRI of the brain and spine might be implemented in the diagnostic workup, e.g., in NEC and atypical lung NEN. An inclusion of those patients into clinical trials should be facilitated and encouraged.

## Figures and Tables

**Figure 1 biology-10-00277-f001:**
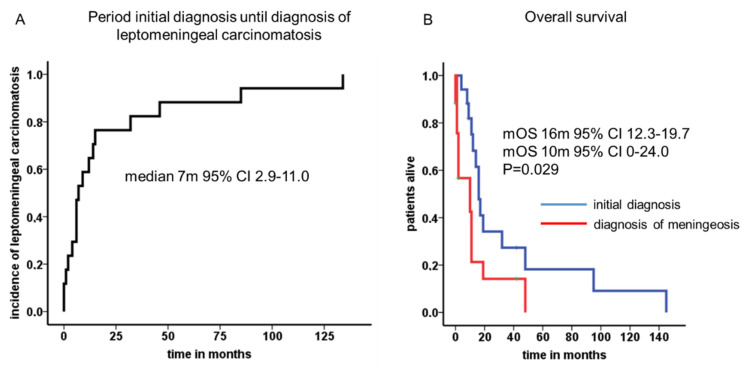
Period of initial diagnosis until diagnosis of leptomeningeal carcinomatosis and median overall survival times. A median latency of 7 months (95% CI 2.9–11.0) was calculated from initial diagnosis to leptomeningeal manifestation (**A**). The cumulative median overall survival reached 16 months (95% CI 12.3–19.7), whereas the median overall survival was 10 months (95% CI 0–24.0) after leptomeningeal carcinomatosis was diagnosed (**B**).

**Figure 2 biology-10-00277-f002:**
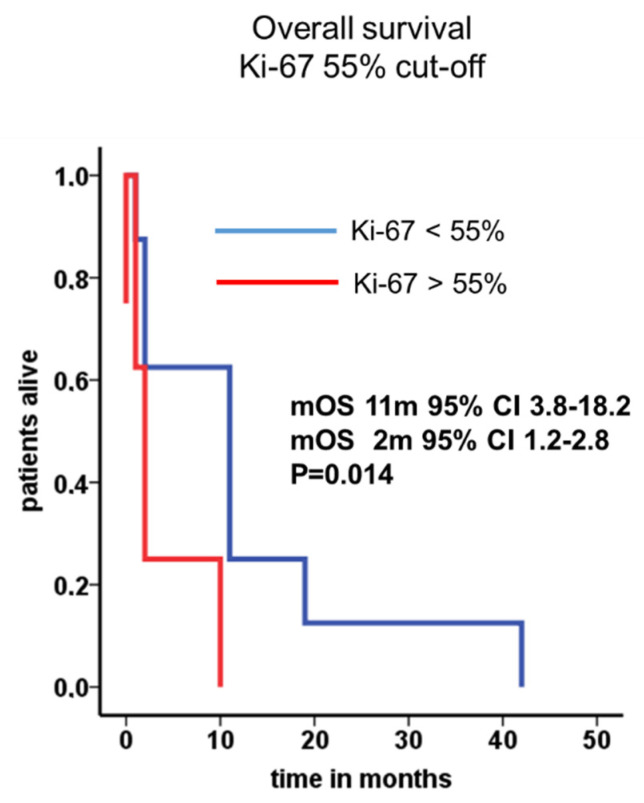
Median overall survival in patients with LC and NEC stratified by Ki-67. Median overall survival in patients with Ki-67 <55% (11 months, blue line) or higher than 55% (2 months, red line) were depicted.

**Figure 3 biology-10-00277-f003:**
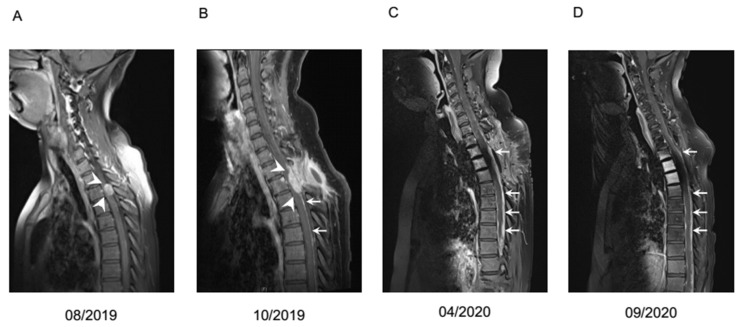
Case example of a patient with leptomeningeal carcinomatosis of a NET G2 with unknown primary, Ki67 20%. Contrast medium enhanced T1 MRI at different timepoints. (**A**) Initial focal meningeal manifestation (white arrowheads) which was locally resected. (**B**) First local recurrence (white arrowheads) with diffuse meningeal enhancement (white arrows), treated with resection and additive irradiation. (**C**) Second recurrence with diffuse nodular meningeal enhancement (white arrows). Liquor analysis showed pleocytosis of 14/µL with 50% tumor cells and signs of blood brain barrier dysfunction. Systemic chemotherapy with carboplatin and etoposide was initiated in April 2020. (**D**) Imaging after four cycles of carboplatin and etoposide revealed slightly regressive tumor manifestations. However, because of progressive neurological deterioration and hematologic toxicity, a switch to treatment with capecitabine and temozolomide was recommended. Deterioration of clinical situation proceeded rapidly, so the patient could not receive the planned chemotherapy and passed away in December 2020.

**Table 1 biology-10-00277-t001:** Summary of patient characteristics. Abbreviations: NET, neuroendocrine tumors; NEC, neuroendocrine carcinomas; CUP, carcinoma of unknown primary; CTx, chemotherapy.

Characteristics	Number of All Patients (%)
**Total**	17
**Mean age at first diagnosis** (years) (range)	55 (27–73)
**Mean age at diagnosis of leptomeningeal carcinomatosis** (years) (range)	57 (27–73)
**Primary tumor localization**	
lung	3 (17.6)
CUP	7 (41.2)
pancreas	3 (17.6)
gastrointestinal tract	2 (11.8)
cervix/prostate	2 (11.8)
**Gender**	
male	9 (52.9)
female	8 (47.1)
**Histology WHO 2010**	
well/moderately differentiated	3 (17.6)
poorly differentiated	14 (82.4)
unknown	0
**Ki-67 index**	
G1 (≤2%)	0
G2 (3–20%)	3 (17.6)
G3 (>20%)	13 (76.5)
<55%	5 (29.4)
>55%	8 (47.1)
unknown	1 (5.9)
**Sites of non-meningeal metastases**	
Brain	9 (53.9)
Liver	9 (53.9)
lymph nodes	10 (58.8)
bone	9 (53.9)
lung/pleural	4 (23.6)
none	1 (5.9)
other	4 (23.6)
**Meningeosis confirmed by**	
Symptoms and radiology	12 (70.6)
CSF cytology	5 (29.4)
**Therapy of LC**	
radiation	11 (64.7)
intrathecal CTx	2 (11.8)
temozolomide-based CTx	5 (29.4)

**Table 2 biology-10-00277-t002:** Symptoms related to leptomeningeal carcinomatosis.

***Symptoms***	***n***	***%***
Headaches	1/17	5.9
Nausea	1/17	5.9
Paraparesis	3/17	17.6
Paresthesia	3/17	17.6
Visual impairment	3/17	17.6
Incontinence	2/17	11.8
None	2/17	11.8
*Others*	*5/17*	*29.4*

Others: Vigilance changes (delirium, somnolent), nystagmus, paralysis of cranial nerves.

**Table 3 biology-10-00277-t003:** Summary of previously reported non-SCLC NEN patients with leptomeningeal carcinomatosis.

Reference	Primary	Histology	Treatment	OS after LC (Months)	Follow-Up after LC (Months)
[25]	Adrenal gland	Pheochromocytoma	NR	NR	NR
[45]	Cervix uteri	NEC	RT, CTx	7.0	7.0
[17]	Cervix uteri	NEC	NR	NR	NR
[16]	Cervix uteri	NEC	BSC	0.5	0.5
[15]	Cervix uteri	Atypical carcinoid	RT	0.3	0.3
[11]	Cervix uteri	NEC	NR	NR	NR
[18]	Colon	NEC	BSC	0.2	0.2
[19]	Intraspinal	Paraganglioma	RT, intrathecal CTx (ThioTEPA), CTx (temozolomide, capecitabine)	NR	36.0
[24]	Intraspinal	Paraganglioma	Surgery, RT	NR	132.0
[24]	Intraspinal	Paraganglioma	Surgery, RT, CTx	NR	30.0
[24]	Intraspinal	Paraganglioma	Surgery	NR	144.0
[46]	Intraspinal	Paraganglioma	NR	NR	NR
[26]	Intraspinal	Paraganglioma	Surgery, RT	4.5	4.5
[47]	Intraspinal	Paraganglioma	Surgery	NR	12.0
[48]	Lung	LCNEC	RT, intrathecal CTx (MTX, dexamethasone)	4.9	4.9
[20]	Lung	LCNEC	RT, CTx (capecitabine)	NR	9.0
[49]	Lung	LCNEC	NR	NR	NR
[50]	Lung	LCNEC	Surgery, RT	NR	12.0
[51]	NR	Carcinoid	NR	NR	NR
[10]	Pancreas	NEC	BSC	1.0	1.0
[22]	Pancreas	NET G1	RT	4.0	4.0
[23]	Pancreas	NET G2	Surgery	NR	0.5
[52]	Pituitary	Pituitary carcinoma	CTx (carboplatin, etoposide)	NR	NR
[13]	Pituitary	Pituitary carcinoma	BSC	0.3	0.3
[53]	Prostate	LCNEC	BSC	1.0	1.0
[14]	Prostate	NEC	Intrathecal CTx (MTX)	NR	NR
[54]	Prostate	SCNEC	Surgery, RT	NR	NR
[54]	Prostate	SCNEC	RT	2.0?	2.0?
[55]	Sinunasal	NEC	RT, CTx	35.3	35.3
[55]	Sinunasal	MiNEN	RT, CTx	4.5	4.5
[55]	Sinunasal	MiNEN	BSC	4.9	4.9
[55]	Sinunasal	SCNEC	RT, CTx	2.8	2.8
[56]	Sinunasal	NEC	NR	NR	NR
[57]	Skin	MCC	Intrathecal CTx (MTX), CTx (ifosfamide), RT	NR	1.0
[58]	Skin	MCC	RT	8.0	8.0
[59]	Skin	MCC	RT, intrathecal CTx	6.0	6.0
[60]	Skin	MCC	RT	6.0	6.0
[21]	Unclear	NEC	Surgery, RT, CTx (temozolomide, endostatin)	NR	10.0
[12]	Bladder	SCNEC	Intrathecal CTx, RT	0.9	0.9

Abbreviations: NR, not reported; NET, neuroendocrine tumor; NEC, neuroendocrine carcinomas; LCNEC, large-cell neuroendocrine carcinoma; SCNEC, small-cell neuroendocrine carcinoma; MiNEN, mixed neuroendocrine non-neuroendocrine neoplasm; MCC, Merkel cell carcinoma; BSC, best supportive care; CTx, chemotherapy; RT, radiotherapy.

## Data Availability

The dataset supporting the conclusions of this article is available on request by contacting the authors.

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
