# Peer review of "Leptomeningeal Carcinomatosis: A Clinical Dilemma in Neuroendocrine Neoplasms"

_biology, 2021, doi:10.3390/biology10040277_

Round 1

Reviewer 1 Report

In their manuscript, Apostodolis and Co-Workers aimed to evaluate the frequency, clinical characteristics, diagnostic steps, treatments and outcomes of Leptomeningeal Carcinomatosis (LP) in patients with NEN (excluding SCLC and LCLC) from four NET Centres.

The identified 17 patients with LC. Confirmed LC was defined as the presence of malignant cells corresponding to the underlying disease in a liquor puncture.

The manuscript is well conducted and well structured.

Major points:

  • What is the frequency of LP in the population? ( I might have missed this figure, included the total number of the NEN patients evaluated)
  • Do the Authors consider appropriate the G1/G2/G3 classification for all NENs? (I.e. typical and atypical lung carcinoids?
  • Given the definition of LC confirmation (CSF cytology), how the percentages of Meningeosis confirmed by non-CSF cytology criteria (Table 1) should be interpreted? Same point in page 4, lines 120 to 123

Minor Point

  • Is mOS “median Overall Survival”?

Author Response

We have attached a point by point reply.

Reviewer 2 Report

Interesting data on a topic not frequently studied. Seventeen patients are presented however following the definition of LC, only 5 fulfilled the criteria set by the authors (positive liquor puncture). Why was liquor puncture not performed in all 17 patients? It should be pointed out that in 12/17 patients there was suspicion of LC. Although the numbers are very low, was there any difference between the 5 patients with positive liquor puncture and the 12 patients with suspicion of LC based on clinical and radiologic findings, respectively?
What are radiological criteria for the diagnosis of LC?
What are the findings on PET/CT (18F FDG and 68Ga DOTA-) in these patients?

Author Response

(The authors gave the same response as above.)
